# The Role of Notch Signaling and Gut Microbiota in Autoinflammatory Diseases: Mechanisms and Future Views

**DOI:** 10.3390/biomedicines13040768

**Published:** 2025-03-21

**Authors:** Vincenzo Giambra, Mario Caldarelli, Laura Franza, Pierluigi Rio, Gaja Bruno, Serena di Iasio, Andrea Mastrogiovanni, Antonio Gasbarrini, Giovanni Gambassi, Rossella Cianci

**Affiliations:** 1Institute for Stem Cell Biology, Regenerative Medicine and Innovative Therapies (ISBReMIT), Fondazione IRCCS “Casa Sollievo della Sofferenza”, 71013 San Giovanni Rotondo, Italy; v.giambra@operapadrepio.it (V.G.);; 2Department of Translational Medicine and Surgery, Catholic University of Sacred Heart, 00168 Rome, Italy; mario.caldarelli01@icatt.it (M.C.); andrea.mastrogiovanni01@icatt.it (A.M.); giovanni.gambassi@unicatt.it (G.G.); rossella.cianci@unicatt.it (R.C.); 3Fondazione Policlinico Universitario A. Gemelli, Istituto di Ricerca e Cura a Carattere Scientifico (IRCCS), 00168 Rome, Italy; 4Department of Emergency Medicine, AOU Modena, 41125 Modena, Italy

**Keywords:** Notch signaling, autoinflammatory, gut microbiota, prebiotics, immunomodulation

## Abstract

Notch signaling is an evolutionarily conserved, multifunctional pathway involved in cell fate determination and immune modulation and contributes to the pathogenesis of autoinflammatory diseases. Emerging evidence reveals a bidirectional interaction between Notch and the gut microbiota (GM), whereby GM composition is capable of modulating Notch signaling through the binding of microbial elements to Notch receptors, leading to immune modulation. Furthermore, Notch regulates the GM by promoting SCFA-producing bacteria while suppressing proinflammatory strains. Beneficial microbes, such as *Lactobacillus* and *Akkermansia muciniphila*, modulate Notch and reduce proinflammatory cytokine production (such as IL-6 and TNF-α). The interaction between GM and Notch can either amplify or attenuate inflammatory pathways in inflammatory bowel diseases (IBDs), Behçet’s disease, and PAPA syndrome. Together, these findings provide novel therapeutic perspectives for autoinflammatory diseases by targeting the GM via probiotics or inhibiting Notch signaling. This review focuses on Notch–GM crosstalk and how GM-based and/or Notch-targeted approaches may modulate immune responses and promote better clinical outcomes.

## 1. Introduction

Discovered over a century ago, the Notch gene was named after the wings of *Drosophila melanogaster*, which, in mutants, take on a notched appearance [1].

The Notch gene belongs to the transmembrane receptor family and participates in one of the most conserved signaling pathways, the Notch signaling pathway, which is important in intercellular communication and control of cell fate. This pathway regulates many important biological processes including cell growth and stem cell differentiation and maintenance during embryonic and adult development [2].

Consequently, abnormalities or dysregulation of Notch signaling are associated with a wide variety of human diseases, such as developmental anomalies and adult cancers. In the past years, Notch has been found to regulate innate and adaptive immunity, providing a basis for Notch-targeted therapeutic approaches in immune disorders [3].

The Notch pathway is at the crossways of innate and adaptive immune systems and may be implicated in the pathogenesis of autoinflammatory diseases.

Autoinflammatory diseases (AIDs) are a heterogeneous group of diseases, which differ from autoimmune diseases and are characterized by recurrent episodes of spontaneous, dysfunctional, non-infectious sterile processes of inflammation [4]. Since the initial cloning of the familial Mediterranean fever gene in 1997, there has been a notable acceleration in identifying novel AIDs. As of 2022, the International Union of Immunological Societies reported a total of 485 inborn errors of immunity, many of which exhibit characteristics typical of autoinflammation [5]. The pathophysiology underlying AIDs is inherently intricate: while certain conditions arise from rare mutations in genes that govern innate immunity, others are polygenic in nature. In such cases, environmental triggers can modify disease manifestation in individuals with a genetic predisposition [6]. Barriers to diagnosing AIDs include limited access to advanced genetic testing and long waiting times for genetic consultations [7]. Most physicians have access to panel testing, but this may not include newly identified genes. One potential solution is research enrollment for exome/genome sequencing and transcriptomic profiling. Wider availability, as technology becomes more advanced and cheaper, may reduce diagnostic delays, increase accuracy, and improve patient outcomes [8].

In 2015, Xu et al. studied the effect of altering Notch signaling on experimentally induced inflammation using both genetic and pharmacological approaches. These methods ranged from loss-of-function deletion of pathway components across lymphoid or myeloid cell divisions to systemic inhibition of all classes of Notch signaling, as well as specific blockading of certain Notch receptors and ligands [9].

Additionally, in macrophages, Notch signaling intersects with other signaling cascades, such as Nuclear Factor kappa-light-chain-enhancer of activated B cells (NF-κB) signaling, in the context of inflammatory responses. Recently, in 2022, Li et al. reported that the lipopolysaccharide (LPS), a ligand for Toll-like receptor (TLR)-4, could upregulate Notch1 expression in macrophages [10]. Notably, TLR4 signaling also promoted Notch target gene transcription, including Notch1 and hairy and enhancer of split-1 (Hes1), suggesting that Notch activation in macrophages is upregulated by TLR4 signaling.

Notch activation is known to induce a proinflammatory response, indicating that pharmacological blockading of this pathway could be a potential therapeutic target for the treatment of numerous inflammatory diseases [11]. These diseases are characterized by aberrant activation of the immune system that leads to systemic inflammation. Although similar to autoimmune diseases, autoinflammation primarily involves the innate immune system [12].

The Notch pathway also interacts with the microbiota, which is composed of various microorganisms living in the body, and it plays a key role in maintaining homeostasis [13,14].

The microbial community living in the intestine, known as the gut microbiota (GM), has been thoroughly studied and has been proven to be involved in many different diseases [15].

The GM consists of bacteria, viruses, and fungi living in the intestinal lumen [14]. Its composition can vary between individuals and even within the same individual over time. There are six main phyla (*Firmicutes*, *Bacteroidetes*, *Actinobacteria*, *Proteobacteria*, *Fusobacteria*, and *Verrucomicrobia*) and their equilibrium has serious consequences on human health [16]. In particular, the GM has been shown to modulate the host’s immune system through different pathways, impacting its development and interacting with both innate and adaptive immunity [17]. The GM initiates innate intestinal immune responses by activating pattern recognition receptors (PRRs), releasing cytokines, and promoting the production of antimicrobial peptides, whereas it contributes to adaptive immunity by influencing the differentiation of CD4+ T cells, B cells, and cells of the lamina propria [18].

Among its functions, the GM regulates the proliferation and differentiation of intestinal stem cells (ISCs) through Notch and Wnt pathways via PRRs [19].

An imbalance in the interactions between the GM and immune system may result in gut dysbiosis, disruption of the intestinal barrier, local and systemic inflammation, and deregulated immune responses [20].

It has been demonstrated that the Notch pathway can mediate the impact of the GM on autoinflammation.

For instance, *Listeria monocytogenes* infection blocks the Notch1/Hes1 pathway, which stimulates the differentiation of intestinal stem cells into goblet cells and leads to diarrhea [21]. Similarly, *Desulfovibrio vulgaris*, which is pathologically upregulated in ulcerative colitis (UC), induces Notch1 signaling in a TLR4-independent manner [22]. Flagellin has been reported to upregulate Notch1 signaling and induce Intelukin-6 (IL-6) production through both NF-κB and the recombination signal binding protein for the immunoglobulin kappa J region (RBP-Jκ) [23].

Conversely, certain beneficial microbes can modulate Notch signaling and promote intestinal health. For instance, *Lactobacillus acidophilus* inhibits the Notch1 pathway, which is beneficial for intestinal mucosal healing, and reduces the loss of goblet cells in UC induced by *Salmonella* infection [24,25]. Similarly, *Helicobacter pylori* activation of nucleotide-binding Leucine-rich repeat-containing 12 (NLRP12) suppresses Notch signaling and leads to decreased intestinal epithelial cell production of inflammatory chemokines, such as monocyte chemoattractant protein-1 (MCP-1) and Macrophage Inflammatory Proteins-1α (MIP-1α) [26].

This interaction underscores the temporal and cumulative effects of microbial modulation of the Notch pathway on gastrointestinal homeostasis and disease.

This narrative review describes the role of the Notch signaling pathway and GM in the pathogenesis and modulation of autoinflammatory diseases. It also examines the potential therapeutic benefits of GM modulation and Notch inhibition, identifies the probiotic and anti-Notch strategies for immune modulation, and explores their impact on clinical outcomes.

The review was performed based on an electronic literature search of PubMed, MEDLINE, and Google Scholar, using keywords including “Notch pathway”, “autoinflammatory diseases”, “inflammation”, and “gut microbiota”. We considered peer-reviewed articles, including both original and review articles, written in English. We selected articles based on their relevance to the subject, study design, approach, and sample size, with studies both on small and large scales. A manual search of articles using references in relevant articles was also conducted to increase the number of studies included in this review.

## 2. The Notch Signaling Pathway

Notch signaling is an evolutionarily conserved pathway that controls various developmental and homeostatic processes in metazoans [27].

In mammals, the Notch pathway consists of Notch receptors, ligands, and signaling effectors [28]. In *Drosophila melanogaster* [29], there is one Notch receptor ortholog, Notch1. Most mammals, however, possess three other Notch receptors: Notch2, Notch3, and Notch4. Notch receptors are type I transmembrane proteins with three major classes of domains: the extracellular domain (NECD), the transmembrane domain (TMD), and the intracellular domain (NICD) [30] (Figure 1).

NICD include the recombination signal binding protein for the immunoglobulin kappa J region (RBPJ) association module (RAM) that mediates NICD binding to RBPJ to activate transcriptional activity; nuclear localization sequences (NLSs) that allow NICD transport into the nucleus to function as a transcription factor; ankyrin repeats (ANKs) involved in protein–protein interactions during Notch signaling complex assembly; a transactivation domain (TAD) that recruits co-activators required to activate the transcription of target genes; and a conserved proline/glutamic acid/serine/threonine-rich motif (PEST) that regulates NICD degradation [31].

In humans and mice, five delta-like ligands bind to the extracellular regions of Notch receptors [32]. The ligands can be divided into serrate-like (Jagged1, JAG1, and Jagged2, JAG2) and delta-like (DLL1, DLL3, and DLL4) families, depending on the presence of a cysteine-rich domain [32]. Notch ligands are also transmembrane proteins with structural resemblance to Notch receptors. JAG1 and JAG2 contain DSL, EGF-like repeats, and cysteine-rich regions in their extracellular domains. In contrast, DLL1, DLL3, and DLL4 have EGF-like repeats homologous to JAG1 and JAG2 in their extracellular domains but lack the cysteine-rich region [33].

The Notch signaling pathways are divided into canonical and non-canonical pathways.

The canonical pathway plays a significant role in cell fate determination and intercellular communication, regulating embryonic development, tissue differentiation, and gene regulation, as well as contributing to both benign and malignant diseases [34]. The Notch signaling pathway involves multiple steps for the maturation and activation of Notch proteins [35] (Figure 2).

Activation progresses from the Golgi apparatus to the cell membrane, where Notch proteins change from single-chain precursors to functional proteins. In response to stimulation by ligands, Notch receptors are activated and associate with the DNA-bound centromere binding factor-1 (CBF1) and the Suppressor of Hairless, Lag-1 (CSL) co-repressor complex to initiate the transcription of downstream target genes [36]. In the Figure 3, we report the steps of the Notch activation process.

In particular, S1 cleavage occurs at the Golgi apparatus, producing a heterodimeric receptor [37] and, when ligands bind to the cell surface, they induce cleavage at the S2 site by disintegrins and metalloproteinases, thus freeing a component (TMD + NICD) known as NeXT [38]. At the S3 site, γ-secretase cleaves NeXT, releasing the NICD. The NICD moves into the nucleus, where it forms a complex with CSL and co-activator proteins, with a subsequent change into a co-activator complex [27]. This shift promotes the expression of Notch target genes, whereas, in the absence of NICD, CSL represses gene expression [39].

Non-canonical Notch signaling modulates other pathways, such as Wingless (WNT)/β-catenin, Janus Kinase/Signal Transducer and Activator of Transcription (JAK/STAT), Phosphoinositide 3-Kinase/Protein Kinase B (PI3K/AKT), and NF-κB through post-translational mechanisms [40]. In malignancy, non-canonical Notch activation has been associated with cell proliferation, apoptosis regulation, and tumor vascularization [41].

Research has associated the Notch signaling pathway with some inflammatory mediators. For instance, Jundt et al., in 2004, demonstrated that increased Jagged1, Notch1, and Notch2 levels correlate with myeloma progression [42]. Furthermore, there is evidence that Notch has an activating role in proliferative signals mediated by IL-6 in the bone marrow niche, leading to enhanced tumor growth [43].

Using a murine model for pancreatic cancer, Maniati et al. observed that the combination of tumor necrosis factor (TNF)-α, basal Notch signaling, and IκB kinase 2—a component of the NF-κB pathway—suppresses the nuclear receptor peroxisome proliferator-activated receptor gamma (Pparg) [44]. Pparg is repressed by Hes1, which upregulates the inflammatory activity of pancreatic tumor cells in an autocrine manner, releasing inflammatory mediators, such as TNF-α, IL-6, and IL-1β. This process induces a positive feedback loop, which further drives the advancement of pancreatic cancer via Notch pathway activation [44].

Notch signaling is essential for sustaining stem cell features and directing cell fate in the intestinal mucosa, while homeostasis is maintained by the balance of epithelial-derived cells. Endogenous Notch activation promotes epithelial repair after inflammatory stress [45]. A study by Kim et al. showed that, in a murine model, Notch activation suppressed intestinal adenoma progression, indicating that Notch may have a tumor suppressor role. These results underline the potential role of Notch signaling in the intestinal microenvironment as a mechanism for tissue repair and cancer prevention [46]. Several studies have also reported the cooperation between the WNT and Notch pathways in chronic inflammatory contexts, such as colitis-associated tumorigenesis [47]. Taniguchi et al. discovered that the co-receptor gp130 activates Yes-Associated Protein and Notch to promote epithelial cell proliferation, providing a new connection between Notch and inflammation [48].

Notch signaling is also modulated in the immune cells, including macrophages, which contribute to carcinogenesis [49]. Macrophage polarization (classically activated, M1, versus alternatively activated, M2) is dependent on environmental signals and is important for the relationship between Notch and inflammatory pathways in cancer development [50].

M1 macrophages respond to microbial stimuli from the TLR pathway by upregulating inflammatory mediators, whereas M2 macrophages express fewer inflammatory factors and play a role in host defense and the resolution of inflammation. Inflammatory mediators vary with disease progression: during the shift from acute to chronic colitis, there is a transition from Th1-Th17 cytokines to a Th2-mediated inflammatory response [51].

Different macrophage functional phenotypes are associated with Notch activation. A study by Outz et al. showed that Notch1 deficiency in macrophages alters the expression of vascular endothelial growth factor receptor-1 (VEGFR-1) and inflammatory cytokines including TNF-α, ultimately resulting in decreased inflammation during the process of wound healing [52]. Furthermore, Notch1 contributes to M1 polarization via the RBP-J-TLR4-IRF8 axis [53].

In 2015, Kueanjinda et al. identified a new function for the protein Numb, a negative regulator of Notch1 signaling, in amplifying the production of proinflammatory cytokines, including TNF-α, IL-6, and IL-12 by macrophages [54]. Notably, in the bone marrow, persistent Notch activation due to Numb loss did not interfere with monocyte differentiation into macrophages [54].

Notch signaling also contributes markedly to inflammation in several pathological conditions where specific receptors and ligands interact with effector inflammatory cells and/or endothelial cells [55,56]. Emerging evidence indicates that Notch signaling is closely associated with inflammatory responses. This interaction is mainly mediated in immune cells by the direct functional role of Notch in several critical cellular processes [55].

Notch-related signaling has been shown to play a role in inflammatory responses and, given the multitude of AIDs, understanding how Notch signaling interacts with these diverse immune disorders is critical.

Altered immune responses with multi-organ involvement and resulting tissue damage are hallmarks of AIDs. This injury is linked to hazard cues, including non-microbial damage-associated molecular patterns (DAMPs) and microbial pathogen-associated molecular patterns (PAMPs), perceived by PRRs. The receptors that belong to this category are TLRs and nucleotide-binding oligomerization domain-containing (NOD)-like receptors (NLRs). These receptors initiate an inflammatory response by inducing the release of proinflammatory cytokines, mainly from innate immune cells, including neutrophils, monocytes, macrophages, and mast cells, upon ligation [57].

Furthermore, some AIDs are recognized as IL-1-mediated diseases and can be effectively treated by anti-IL-1 agents, suggesting that targeted therapeutics could be successful in managing these diseases [58]. Hyperactivation of the innate immune system and uncontrolled production of proinflammatory cytokines (IL-1β, IL-18, TNF, type-1 interferons) are characteristic features of autoinflammation [59]. The NLR family pyrin domain-containing 3 (NLRP3) inflammasome, an intracellular molecular sensor that stimulates inflammation and causes programmed cell death, can release IL-1β and IL-18. Nuclear factor-kB (NF-kB) signaling is another major pathway involved in autoinflammation [60]. This pathway is governed by multiple post-translational mechanisms and is essential for triggering inflammation by enhancing the expression of proinflammatory chemokines and cytokines, including IL-1, IL-6, and TNF, as well as by modulating inflammasome signaling [61].

As previously discussed, Notch signaling interacts bidirectionally with the GM: on the one hand its composition can influence the Notch pathway, while on the other, Notch signaling can drive shifts in GM composition [62]. In the following sections, we will discuss GM composition and its complex crosstalk with Notch signaling in the context of autoinflammatory diseases.

## 3. Gut Microbiota and Autoinflammatory Diseases

The GM has both a direct and indirect effect on immune function: a healthy GM, for instance, helps maintain a healthy gut barrier, preserving the normal mucus layer, as in the case of *Akkermansia muciniphila*, or avoiding colonization from harmful pathogens, which has also been associated with *Akkermansia muciniphila*, but also with *Escherichia coli H22*. The mechanism through which *Akkermansia muciniphila* is able to prevent dysbiosis is linked to its ability to produce short chain fatty acids (SCFAs), which in turn are also linked to metabolic health and direct immune modulation [63]. Other bacteria producing SCFAs have indeed also been associated with health benefits: *Bifidobacterium*, for instance, promotes both immune system maturation and the integrity of the intestinal barrier, similarly to *Lactobacillus*, *Bacteroides*, and other strands [64].

The role of the GM on the innate immune system is highly relevant in the context of autoinflammatory diseases [65,66]. Research has focused on the relationship between GM and autoinflammatory conditions, uncovering various underlying mechanisms [67].

Notably, the interaction between the innate immune system and the GM is particularly complex [68]: the GM is capable of influencing TLR signaling, the myeloid differentiation primary response protein (MyD88), tumor necrosis factor (TNF), receptor-associated factor 6 (TRAF6) and NF-κB essential regulator (NEMO), receptor-interacting serine/threonine-protein kinase 1 (RIPK1), FAS-associated death domain protein (FADD) and caspase-8, and NOD-containing protein 2 (NOD2) through interaction with the epithelial cells of the intestine [69]. The GM is also responsible for the maturation of myeloid cells and of macrophages. For instance, in a study by Erny et al., it was observed that microglia in germ-free mice present numerous defects and do not develop correctly [70]. The GM also has an effect on innate lymphoid cells. While it does not appear to affect their development, innate lymphoid cells fail to function correctly in the absence of commensal microorganisms [71]. Additionally, a healthy GM has been linked to the proper monocyte functions. For instance, it was observed that GM modulation, promoting the presence of *Akkermansia muciniphila*, can increase monocyte IFN-I production in the context of cancer [71,72]. Neutrophils are another component of the innate immune system which has a complex crosstalk with the GM: they are influenced in their development and function and, on the other hand, they contain the GM population [73]. In particular, components of the GM can promote the proinflammatory activity of neutrophils, through the activation of the TLR-Myd88 pathway; on the other hand, SCFAs exhibit the capacity to inhibit their overactivation [73]. Neutrophils produce IL-1β, a crucial player in AIDs, and the GM can also modulate this aspect of neutrophil activity: in murine models, the development of IL-1β is impacted by the GM and can be modulated through the diet [74]. GM modulation has proven to offer some potential in AIDs also through other methods, such as fecal microbiota transplantation [75].

The importance of the GM in the pathophysiology of autoinflammatory disorders has been extensively studied. In a study by Manukian et al., individuals with familial Mediterranean fever exhibited high levels of circulating antibodies against different GM components, particularly *Bacteroides*, *Parabacteroides*, *Escherichia*, *Enteroccocus*, and *Lactobaccilus* [76]. It is worth noting that *Enterobacter*, *Klebsiella*, and *Ruminococcus gnavus* were found to be elevated in this group of patients and these bacteria taxa have all been linked to increased inflammation. However, when patients were treated with colchicine, their GM composition changed in favor of taxa with anti-inflammatory properties, such as Faecalibacterium and Roseburia [77].

The inflammasome has become one of the key players in the pathogenesis of AIDs [78].

Dysregulation of inflammasome function has been recognized as a crucial element in AID pathogenesis. Inflammasomes are specialized intracellular sensors that respond to a variety of stress signals and pathogens. Studies have demonstrated that GM changes can modulate inflammasomes, specifically the NLRP3 inflammasome. In murine models, dietary modifications reduce the activity of the NLRP3 inflammasome. These data imply that dietary interventions could modulate inflammasomes and inflammatory processes related to AIDs [74].

CD103^+^ dendric cells, as well as IL-17-producing γδT cells, are promoted by the GM and are influenced by GM composition during differentiation. These γδT cells play a role in the pathophysiology of autoinflammatory diseases (AIDs). Research in murine models has demonstrated that dysbiosis can impair CD103^+^ dendritic cells and subsequently alter the activation of IL-17-producing γδT cells [79].

As discussed above, AIDs are characterized by the activation of the innate immune system and the Notch pathway acts at the intersection of the adaptive and innate immune system [80]. The GM has been shown to modulate Notch signaling through the MyD88 pathway in zebrafish [74]. Conversely, Notch signaling has been shown to influence GM composition, particularly by reducing proinflammatory bacteria, such as *Enterococcus* and *Escherichia-Shigella*, while promoting the presence of SCFA-producing bacteria [81]. In the following section, we will explore the role of the Notch pathway in autoinflammatory diseases and the potential mediating role of the GM.

## 4. Notch Pathway, Gut Microbiota, and Autoinflammation

Recent evidence has linked Notch signaling to both innate immunity and inflammation. Active Notch signaling has been detected in various inflammatory diseases, such as rheumatoid arthritis, systemic lupus erythematosus, and some autoinflammatory diseases [82].

Notch signaling in T cell development and regulation is an active area of investigation. In research by Hoyne et al., Notch signaling promotes a regulatory phenotype in murine peripheral CD4^+^ T cells interacting with antigen-presenting cells (APCs) that overexpress the human Jagged1 [83]. Notch signaling was also shown recently to be required for polarization of several T helper cells, including Th1, Th17, and Th2 cells, and the regulatory T cell (Treg) population [11]. This has prompted speculation that Notch acts as an unbiased amplifier of polarizing signals, e.g., cytokines, instead of playing an instructive role in the T helper polarization process [84].

Conversely, Notch signaling plays a pivotal role in the differentiation of cytotoxic (CD8^+^) T cell effectors and in T cell activation processes, including the enhanced production of IL-2, a critical factor in T cell proliferation [85].

Loss-of-function experiments have emphasized the key role of Notch signaling in the context of innate immunity. Recent data have revealed it mediates the proinflammatory activation of in vitro-obtained primary human blood monocyte-derived macrophages. For example, proinflammatory stimuli can activate DLL4 expression through a TLR4-NFκB-dependent pathway [86].

Due to the profound impact of the Notch pathway on inflammatory responses, the scientific community is evaluating the association between Notch signaling and AIDs. Although this is a rapidly expanding field, at present only a handful of AIDs connected to changes in the Notch signaling pathway are known. Despite not falling under the category of genetic disorders, studies have associated AIDs with inherited polymorphisms in proteins including γ-secretase and other Notch-signaling-related proteins [87]. These polymorphisms could alter the response of the inflammasome to various stimuli (such as hormones, tobacco, and adipokines associated with insulin resistance and obesity). As mentioned earlier, GM seems to participate in the pathogenesis of these diseases, indeed adding to the complexity of environmental, genetic, and microbial interaction in AIDs (Figure 4).

Notch signaling can influence AIDs in different ways, which may vary slightly between diseases, but are generally similar. For instance, alterations in the function of γ-secretase have been linked to different AIDs [88]. The γ-secretase complex is a diverse transmembrane protease composed of presenilin catalytic and cofactor subunits, including presenilin enhancer-2 (PSENEN), nicastrin, and anterior pharynx defective 1 (APH1). These subunits are encoded by the genes PSEN1/PSEN2, PSENEN, NCSTN, and APH1A/APH1B, respectively [89]. While it is not directly and only involved with Notch signaling, it is responsible for cleaving more than 140 type I membrane proteins, including cadherins and Notch [89]. This suggests that γ-secretase and its downstream signaling pathways, including Notch, contribute to epithelial remodeling and the chronic inflammation observed in autoinflammatory skin diseases. Also, γ-secretase complex inhibition, a key player of Notch signaling, decreases IL-17, IFN-γ, and CD4+ T cell differentiation into Th17 cells, inflammatory mediators of CD4+ T cells, suggesting a role as a therapeutic strategy particularly in the context of AIDs affecting the intestine. In a paper by Kar et al., it was observed that γ-secretase activation was influenced by GM composition: modulation through the administration of *Bifidobacterium bifidum* and *Lactobacillus salivarius* reduced its activation, which could have interesting consequences in many pathological conditions [90].

Cytokines also play a key role in Notch activation. For example, IL-1β and TNF-α, two major inflammatory cytokines, are well-established Notch activators [91]. High levels of inflammatory cytokines, such as TNF and IL-1β, are released during innate immune responses and inflammation—key defense mechanisms against numerous pathogens. In cases of hyperinflammation and autoimmunity, aberrant regulation of these cytokines can have deleterious and pathogenic effects [82]. One example is that TNF is important in the pathogenesis of rheumatoid arthritis (RA) and a confirmed drug target for RA. Notch pathway activation is characterized by TNF-induced nuclear translocation of NICD and its expression of Notch1, Notch4, and Jagged2, and these are the hallmarks of RA synovial fibroblasts [92]. Crawford et al. (2009) also showed that deleting Notch3 or inhibiting its signaling in a mouse model prevented joint damage in inflammatory arthritis [93]. It is worth noting that TNF, in the context of RA, also influences the composition of the GM: patients who are undergoing therapy with anti-TNF drugs can experience a partial restoration of their GM, particularly experiencing a reduction of Euryarchaeota, which was directly linked to disease activity [94]. A similar mechanism may occur in CAPS. Ottaviani et al. (2010) reported that IL-1β induces expression of the Notch target gene Hes1 in chondrocytes via activation of the receptor Notch1, implying that, like TNF, IL-1β is also a Notch activator [95]. Interestingly, production of IL-1β can also be influenced by the GM: in murine models, for instance, it was observed that its secretion is promoted during infections by the commensal GM, further highlighting the interaction between inflammation, Notch, and the GM [96].

TGFβ also has Notch-activating potential, as shown by studies demonstrating that TGFβ directly induces Hes1 expression in diverse cell types [97], and it also has the ability to regulate the GM: interestingly, the activity it exerts on the GM is part of a crosstalk that influences TGFβ activity in a loop communication, which also impacts Notch signaling [98].

Overall, the role of the GM in inducing and interacting with Notch-activating cytokines is very interesting, as it further highlights the interaction between Notch and GM composition: in a paper by Singh et al., it has been discussed that sulfate reducing bacteria are able to promote the expression of IL-1β and other cytokines, which in turn induce the activation of Notch signaling [22]. This observation offers interesting prospectives on the possibility of modulating Notch signaling in the context of a number of diseases.

In addition, Notch signaling and Hypoxia-inducible factor (HIF)-1α are crucial for both physiological and pathological homeostasis. Hypoxia potentiates Notch signaling and promotes the recruitment of HIF-1α to the NICD complex [99]. Hypoxia also determines changes in the GM, which can in turn impact Notch signaling: HIF-1α is a key regulator of GM homeostasis and helps maintain a healthy microbiota, particularly promoting the expression of SCFA-producing bacteria, showing that its role in modulating Notch signaling may be more nuanced than what could appear at first glance [100]. SCFAs are, indeed, able to modulate the expression of Notch, blocking or promoting its activation in different settings [101]. Moreover, it has been shown that the miR-497-195 cluster can also induce angiogenesis by maintaining endothelial Notch and HIF-1α activity [102].

Non-coding RNAs (ncRNAs) have also been studied in the context of AIDs, and Gu et al. (2023), for instance, observed that small ncRNAs, such as miRNAs, and large lncRNAs are important regulators in AIDs [103] and can have a direct impact on Notch signaling, as in the case of miR-23b, which also induces Th1/Th17 cell response. IL-17- and IFN-γ-expressing T cells increased after transfection of CD4+ T cells with a miR-23b inhibitor [22]. This immune activation, along with the loss of miR-23b’s immunomodulatory properties, may lead to excessive Notch pathway activation and inflammatory cell proliferation, reinforcing its potential role in the pathogenesis of specific AIDs [22]. GM is influenced by the activity of ncRNAs, too, once again highlighting that the mechanisms through which Notch is regulated are often more complex than what could appear at first glance [104].

Hildebrand et al. (2018) demonstrated that TLR signaling promotes the expression of the Notch receptor ligand delta-like 1 (DLL1) and activates Notch signaling in human blood-derived monocytes. The induction of DLL1 by TLR activation occurs indirectly, mediated by cytokine stimulation and autocrine signaling via cytokine receptors, which activate the STAT3. Furthermore, their findings reveal a positive feedback loop between Notch signaling and the Janus kinase (JAK)/STAT3 pathway in vitro, facilitated by Notch-enhanced IL-6 production [105]. Once again, this Notch regulatory mechanism can also be modulated by the GM, with complex implications: TLRs, for instance, are able to recognize harmful GM components, thus tying dysbiosis to Notch activation, but on the other hand, SCFAs are also implicated in modulating TLRs, in a positive sense in this case, overall highlighting that the GM plays a critical role in Notch modulating [106].

### 4.1. Pyoderma Gangrenosum and Acne Spectrum Disorders

Pyogenic arthritis, pyoderma gangrenosum and acne (PAPA) syndrome, Proline-Serine-Threonine Phosphatase-Interacting Protein 1.-associated myeloid-related proteinemia inflammatory (PAMI) syndrome, pyoderma gangrenosum, acne and purulent acne hidradenitis (PASH) syndrome, and pyogenic arthritis, pyoderma gangrenosum, acne and purulent hidradenitis syndrome (PAPASH) are all autoinflammatory diseases that result in persistent inflammation and tissue damage [107]. Besides the mechanisms previously discussed, mutations in the Protein O-Fucosyltransferase 1 and Protein O-Glucosyltransferase 1. genes, which encode GDP-fucose protein O-Fucosyltransferase 1 and protein O-glucosyltransferase 1—two proteins involved in Notch signaling—have been identified in patients with Hidradenitis Suppurativa (HS) and Dowling-Degos disease [1]. Dowling-Degos disease is an autosomal dominant skin disorder characterized by flexural hyperpigmentation that may occur independently or in association with HS [11].

The GM, in this group of disease, appears to be involved in particular in the γ-secretase pathway and specific microorganisms have been associated with the disease: *Ruminococcus gnavus* and *Clostridium ramosum* have been associated with the onset of HS, while the skin microbiota also appears to be affected [108].

### 4.2. Behçet’s Disease

Behçet’s disease (BD) is a multisystem polygenic autoinflammatory disorder of unknown cause, mainly characterized by recurrent episodes of genital ulcers, oral aphthous ulcers, and uveitis [109]. The pathogenesis of BD is characterized by a highly complex genetic background. In addition to the well-established role of the HLA-B*51 allele as a dominant genetic susceptibility factor, genome-wide association studies (GWASs) and targeted deep resequencing of specific immune response-related genes have identified associations between BD and rare variants in several genes, including TLR4, NOD2, MEFV (innate immunity regulator), IL-10, ERAP1 (endoplasmic reticulum aminopeptidase 1), STAT4 (signal transducer and activator of transcription 4), IL-23Receptor, and IL-12RB2 (interleukin-12 receptor subunit beta 2) [110]. BD exhibits both autoimmune and autoinflammatory characteristics. It is strongly associated with the HLA-B51 gene and is marked by hyperactivation of Th1 and Th17 cells [111]. These cells produce increased levels of several inflammatory substances, such as cytokines, like IL-18, INF-γ, IL-2, and IL-12, which contribute to the symptoms of BD [112].

Ciccarelli et al. suggest that BD has autoinflammatory features, most notably pathogenetic intrinsic neutrophilic hyperactivation. This heightened activity is probably related to greater production of reactive oxygen species and heightened phagocytosis and chemotaxis [113]. These inflammatory responses are characterized by increased levels of proinflammatory cytokines such as IL-1, IL-6, IL-8, and TNF. This is an important insight into BD as an autoinflammatory disease and highlights the fundamental role played by innate immunity and inflammatory processes in many diseases [114].

Qi et al. examined the involvement of Notch signaling factors in BD patients with and without signs of active uveitis, a typical manifestation of the disease [115]. They found that Notch pathway activity in patients with active uveitis is associated with enhanced Th17 responses. Notably, the GM has been implicated in the Th17/Treg imbalance. A decrease in *Clostridium* populations directly reduces SCFAs, which are essential for maintaining Th17/Treg balance [116]. In this patient group, *Succinivibrionaceae* are a common component of the GM, a trait also observed in other autoinflammatory diseases [117].

As demonstrated by Ma et al. (2021), there is a notable elevation in the signal transducer and activator of transcription 3 (STAT3) phosphorylation levels in patients diagnosed with BD [116]. STAT3 regulates the Notch pathway via a positive feedback loop, suggesting that modulating Notch signaling could be a therapeutic strategy for BD [105].

### 4.3. Inflammatory Bowel Diseases

Inflammatory bowel diseases (IBDs) are non-infectious chronic inflammatory diseases that typically affect the epithelium of the gastrointestinal system [118]. IBD is a chronic, progressive, and relapsing condition. Immune dysregulation and inflammation significantly impact patients’ quality of life [119]. These disorders include Crohn’s disease (CD), ulcerative colitis (UC), indeterminate colitis (IC), and unclassified colitis (IBD-U), in addition to other non-infectious inflammatory conditions [120].

Notch signaling is important for intestinal homeostasis and ISC survival [121]. Within the context of IBD, dysregulated activation of Notch has been associated with upregulation of the HES1 transcription factor in human colonic cell lines [122]. This is associated with elevated differentiation of secretory cell lineages, resulting in an impaired mucus barrier and progression to chronic colitis.

Kuno et al. (2021) discovered that the expression of mRNA for Olfactomedin 4, a marker of ISC, is upregulated by TNF-α and the Notch pathway in patients with IBD [123].

In addition, the Notch signaling pathway has been documented as a pivotal factor in preserving the integrity of tight junctions and adherent junction proteins in murine models. Ahmed et al. demonstrated that during infection with *Citrobacter rodentium*, the absence of Notch signaling impairs the function of tight and adherent junctions. This impairment can lead to increased epithelial permeability, resulting in greater exposure of luminal contents to the immune system and promoting inflammation [124].

Fibrosis is one of the most crucial complications of CD, and the Notch signaling pathway is a critical mechanism for the fibrogenic process, including epithelium mesenchymal transition (EMT) and fibroblast senescence [125].

Recent studies have shown that DLL4 interacts with Notch4 to activate MET transcription factors in colonic epithelial cells. This highlights the complexity of Notch signaling in CD, as it can promote both fibrogenesis via EMT and tissue regeneration via MET. These mechanisms can represent potential therapeutic targets for the control of fibrosis in CD [125].

Ortiz-Masiá et al. (2016) found that M1 but not M2 macrophages promote Notch signaling in epithelial cells by inducing Jag1- and delta-like canonical Notch ligand 4 (Dll4) [126]. This leads to increased HES1 protein levels and Inhibitor of Apoptosis Protein (IAP) activity. In chronic CD patients, the abundance of M1 macrophages correlates with Notch signaling and enterocyte differentiation markers, suggesting that macrophages contribute to the impaired Notch signaling and enterocyte differentiation seen in this disease. In the mucosa of chronic CD patients, an abundance of M2 macrophages is associated with reduced Notch signaling and impaired enterocyte differentiation [126]. The role of the GM in IBDs has long been discussed. While some researchers previously suggested that GM changes result from IBD, it is now widely accepted that the GM actively contributes to IBD pathogenesis [127]. *Bifidobacterium longum*, *Eubacterium rectale*, *Faecalibacterium prausnitzii*, and *Roseburia intestinalis* are reduced in IBD, whereas *Bacteroides fragilis* is increased. *Akkermansia muciniphila* has also been associated with the development of IBDs, even though there is contrasting evidence on its role [128]. A particularly interesting aspect for the purpose of this review is the role of the GM in the interaction between Notch signaling and IBD development and progression: in a study conducted on mice, Liu et al. observed that administrating *Lactobacillus rhamnosus* dm905 and *Lactococcus lactis* had a positive effect on inflammation in IBD, which was at least in part due to the reduction in the inflammatory activity of the Notch pathway [129].

Furthermore, the role of immune activation in the mechanisms and pathophysiology of irritable bowel syndrome (IBS) is currently believed to be the most relevant [130]. This linkage provides a potential mechanism for Notch signaling pathway abnormalities in the pathogenesis of IBS. In agreement, other studies have noted that aberrant Notch signaling initiates an inflammatory response that drives disease more rapidly in IBS [131]. This suggests that Notch signaling pathway inhibition may be a novel therapeutic approach for IBS therapy [132].

### 4.4. Autosomal Dominant Autoinflammatory Disease

Among the autoinflammatory diseases, it is imperative to mention recurrent fevers, and particularly cryopyrinopathies.

Mutations in the NLRP3 gene are responsible for autosomal dominant autoinflammatory disease (NLRP3-AID), and hearing loss (HL) may be the primary or even the only significant symptom of NLRP3-AID. NLRP3 was identified because of its association with autosomal dominant autoinflammatory diseases (NLRP3-AIDs), also known as CAPSs (cryopyrin-associated periodic syndromes) [133]. These disorders include a range of inflammatory symptoms, including urticaria, conjunctivitis, myalgia, arthralgia, fever, headache, and fatigue, with varying degrees of severity. In a study by Yao et al., it has been observed that GM modulation appears to be able to reduce NLRP3 hyperactivation through the induction of Tregs [134]. The mechanisms through which Notch influences this group of diseases have been discussed previously, but it is worth noting that studies on how the interaction between Notch and the GM are showing some promise in further understanding the complex underlying crosstalk [135]. One particularly promising area of research is the composition of the oral microbiota in the context of AIDs: the presence of specific strands, such as *Streptococcus salivarius*, have been linked to reduced cytokine load, particularly those defined as Notch-activating proinflammatory cytokines, but research is still progressing [136].

## 5. Notch-Targeted Drugs and Future Perspectives

Notch signaling differs from other pathways by operating through stoichiometric interactions between its components [137]. Although phenotypic responses to NICD overexpression vary, the downstream effects of Notch activation are typically dose-dependent [138]. This suggests that complete pathway inhibition may not be necessary for therapeutic efficacy [139].

Notch signaling triggers cell-specific responses at different times, and systemic inhibition could affect multiple tissues. To maximize therapeutic effectiveness, it is crucial to calibrate the level and timing of pathway inhibition to manage disease progression while minimizing adverse effects [11]. The roles of different Notch receptors and ligands in driving specific outcomes remain largely unexplored. Due to their accessibility to circulating therapeutic agents and their role in inflammatory diseases, these transmembrane proteins represent promising therapeutic targets.

However, as observed in cancer research, the potential disadvantages of Notch inhibition need to be carefully taken into account. For instance, the loss of Notch-dependent differentiation of gastrointestinal precursor cells into epithelial cells leads to an imbalance with an excess of secretory goblet cells and gastrointestinal toxicity [140,141].

According to the mechanism employed, Notch-targeted drugs can be classified as illustrated in Table 1.

One potential approach involves the inhibition of γ-secretase. Gamma-secretase inhibitors (GSIs), including DAPT (N-[N-(3,5-difluorophenacetyl)-L-alanyl]-S-phenylglycine t-butyl ester) and DBZ (dibenzazepine), have emerged as promising therapeutic agents for diseases where Notch signaling plays a key role in pathogenesis or progression [142].

The initial clinical trial involving MK-0752 was conducted over a decade ago, with participants diagnosed with T cell acute lymphoblastic leukemia/lymphoma (T-ALL). Subsequent studies have examined the efficacy of MK-0752 in treating refractory central nervous system, pancreatic, and both metastatic and early-stage breast cancer [11].

A phase 3 trial of the GSI semagacestat was terminated before completion due to a lack of clinical efficacy and safety concerns [143]. The adverse events observed in this trial included infections, skin reactions, and cancers, which suggested that these issues were related to on-target inhibition of Notch signaling.

Similarly, gastrointestinal and dermatologic adverse effects emerged as the main reasons for treatment discontinuation in a phase 2 study aimed at evaluating the safety of the GSI avagacestat in Alzheimer’s disease [144].

The adverse events determined by GSIs generally depend on their non-selective blockage of the overall Notch pathway [140]. The scientific community has proposed the employment of “Notch-sparing” substrate-selective GSIs in order to minimize the toxicity of Notch-inhibiting treatments, particularly related to gut epithelium proliferation and maturation [145].

Novel approaches are currently of particular interest in the field of oncology.

An example is the small-molecule ADAM inhibitor INCB7839, which underwent early-phase clinical trials for solid tumors and breast cancer [146]. Although trials did report mild adverse events (e.g., asthenia and nausea) when the compound was applied as a monotherapy, they also showed significant toxic effects, such as deep vein thrombosis.

Another approach involves CB-103, which disrupts the assembly of the transcriptional complex [147]. This predicted binding site occupies the same pocket as the RAM domain of NICD. CB-103 has demonstrated effectiveness in preclinical trials and, notably, does not appear to cause the gastrointestinal side effects commonly linked to GSIs.

A more elegant and specific approach compared to Notch inhibition is the modulation of its effectors with monoclonal antibodies (mAbs), like Brontictuzumab or Tarextumab [27,148]. Compared to conventional pharmacotherapy, monoclonal antibodies offer several advantages, including increased potency, less frequent dosing regimens, and enhanced specificity towards the target. MAbs are also generally well tolerated with off-target effects such as hypersensitivity reactions tending to occur less commonly than target-related adverse events [149]. However, in a phase 1 study by Ferrarotto et al., a potential gastrointestinal toxicity of Brontictuzumab has been described as an on-target effect of Notch1 inhibition [150]. Similarly, Tarextumab, a cross-reactive antibody inhibiting both Notch2 and Notch3, is well tolerated at lower doses, whereas it may cause nausea, vomiting, diarrhea, and fatigue at higher doses [151]. The implication is that mAbs require accurate dosing and monitoring, as well as conventional treatments. MAbs targeting Notch are based on the IgG scaffold, with distinct domains for antigen binding and effector functions [11]. These modifiable structures have been employed to engineer advanced therapies, which offer prolonged circulation and targeted cell destruction through immune mechanisms, such as antibody-dependent cellular cytotoxicity, antibody-dependent cellular phagocytosis, or complement-dependent cytotoxicity [152].

Due to Notch signaling function in inflammation and increasing interest in combinations to treat inflammatory diseases, studies have been performed.

Real et al. demonstrated that the combination of the GSI dibenzazepine with the glucocorticoid dexamethasone leads to significant improvement in survival in a mouse xenograft model of glucocorticoid-responsive T cell acute lymphoblastic leukemia (T-ALL) [153]. Administration of the glucocorticoid dexamethasone in combination with GSI reduced lethal gut toxicity of the GSI, and induced apoptosis in T-ALL xenografts more effectively than either agent alone. GSI has been demonstrated to potentiate glucocorticoid receptor (GR) activity, resulting in increased GR-mediated toxicity via a synergistic effect [154]. Importantly, dexamethasone mitigates GSI-induced gastrointestinal toxicity; however, mechanistic details are unclear. Moreover, it has been shown that the Notch1 inhibition-mediated efficacy of brontictuzumab is potentiated by dexamethasone due to its synergistic effect [154].

Huang et al. (2016) emphasized that the most promising clinical success of mAbs in a Notch context has been observed with the combination of the Notch ligand delta-like 4 (DLL4) and vascular endothelial growth factor (VEGF) blockade [155]. This assertion is substantiated by their preclinical studies demonstrating the efficacy of enoticumab in combination with the anti-VEGF IgG1 Fc-fusion (aflibercept) across various cancer models [156]. ABL001, a bispecific antibody targeting DLL4 and VEGF, has demonstrated a stronger biological activity than DLL4 or VEGF-targeting mAbs alone. Among the related adverse events, systemic and pulmonary hypertension, asthenia, headache, and anemia have been reported, although ABL001 is generally more tolerated than other mAbs and has no dose-limiting toxicity [157,158].

**Table 1 biomedicines-13-00768-t001:** Key regulators of the Notch signaling pathway and their mechanisms of action. The table compares the classes of compounds targeting Notch signaling, such as γ-secretase inhibitors, inhibitors of transcriptional complex formation, ADAM metalloprotease inhibitors, and monoclonal antibodies. Currently, there is an extensive investigation of these molecules in the therapeutic setting to exploit their potential in different diseases, mainly in the context of autoinflammatory diseases and cancer.

Molecules	Mechanism of Action	References
DAPT	Inhibition of γ-secretase	[142]
Dibenzazepine	Inhibition of γ-secretase	[142]
MK-0752	Inhibition of γ-secretase	[11]
Semagacestat	Inhibition of γ-secretase	[143]
INCB7839	ADAM inhibitor	[146]
CB-103	Inhibitors of formation	[147]
Brontictuzumab, Tarextumab	Monoclonal antibodies	[27,148]

An intriguing study by Qi et al. (2014) investigated the potential association between Notch signaling and BD, with a particular focus on patients with and without active uveitis, a prevalent manifestation of BD [115]. The study’s findings indicated that Notch pathway activation led to an enhanced Th17 response in patients with active uveitis. Furthermore, the administration of γ-secretase inhibitors led to a substantial reduction in the expression of key inflammatory mediators (IL-17 and IFN-γ), as well as an impact on the differentiation of naïve CD4+ T cells into Th17 or regulatory T cells. This was indicated by a decrease in STAT3 phosphorylation, a hallmark of the BD phenotype [115].

Furthermore, several miRNAs have been shown to regulate the Notch signaling pathway. In BD, the activation of the Notch pathway has been associated with a decrease in the expression of miRNA-23b [115]. In patients with BD, lower levels of miR-23b have been implicated in the activation and amplification of Th17 and Th1 cell functions, as well as the Notch signaling pathway. Inhibition of the Notch signaling pathway has been shown to selectively suppress the Th17 response [159]. Research indicates that members of the NF-κB family regulate miR-23b transcription through the Act-1/IL-17 signaling pathway, establishing a positive feedback loop among miR-23b, Notch signaling, and IL-17 [103].

As for PAPA syndrome and other autoinflammatory disorders with skin involvement, in a study by O’Sullivan et al., where the γ-secretase inhibitor niragacestat was administered, 71% of individuals experienced adverse skin toxicities [160]. Additionally, 53% developed follicular and cystic lesions with surrounding inflammation in areas like the axilla and inguinal regions, resembling hidradenitis suppurativa. The resolution of these lesions upon the discontinuation of niragacestat suggests a pivotal role for γ-secretase in the pathogenesis of follicular inflammation [161].

In the context of IBD research, the role of Notch in the gastrointestinal tract has been examined using GSIs or RBP-J genetic modulation [162]. GSIs induce a shift in enterocyte progenitors to secretory cells, activating Atoh1/Math1 and downregulating Hes1. A comparable effect is observed in RBP-J knockout [163]. The observed changes are primarily attributable to Notch1 inhibition, as blocking Notch2 did not affect intestinal morphology. In spite of data supporting the proinflammatory role of Notch in IBD, a non-selective blockage of this pathway may impair the intestinal epithelium integrity, and adverse events may overwhelm the positive effects at this level. In fact, Notch-1 expression regulates innate and adaptive immune responses in the gut [164], and its complete absence has been associated with a higher disease severity in experimental models of colitis [11]. For this reason, further research assessing the benefits and risks of Notch manipulation in IBD is needed.

The investigation of Notch inhibition in CAPS has received attention, as contemporary guidelines advocate for the utilization of IL-1 inhibitors, which have proven to be efficacious and well tolerated by patients [165]. In addition, the Notch signaling pathway has been implicated in the regulation of chronic inflammation mediated by the NLRP3 inflammasome, a protein complex involved in autoinflammatory diseases [166].

It would be beneficial to conduct studies aimed at developing new drugs that target the Notch pathway, as this may provide a novel therapeutic target for conditions such as CAPS. Targeting Notch could act upstream of IL-1 and NLRP3, potentially offering a way to modulate the inflammatory response more effectively and potentially addressing some of the limitations of current treatments.

Concerning GCA, Piggott et al. (2011) demonstrated that the inhibition of Notch signaling with a GSI or soluble Jagged1 led to a reduction in vascular inflammation in a humanized mouse model [167]. This blockade suppressed Th1 and Th17 cell activity by disrupting communication between CD4^+^ T cells and endothelial or smooth muscle cells expressing Jagged1 [168]. Furthermore, the study identified a correlation between impaired Notch4 signaling and compromised Treg function, thereby promoting vascular inflammation [169].

The importance of Notch signaling in inflammatory diseases is a relatively recent discovery, and most of the available data still come from preclinical studies.

Despite some encouraging results, in certain cases Notch-targeted therapies have failed to meet the expectations of the scientific community, due to concerns about their toxicity, the low affinity of antibody-drug conjugates, and the activation of alternative pathways [63].

Among the strategies to increase the efficacy of Notch-based treatments, pulsed Notch inhibition, high selectivity, and combination therapies have been proposed [170].

Furthermore, the design of tailored treatments for patients with inflammatory disorders requires multi-OMICS studies and the integration of molecular and clinical data for a better comprehension of the pathogenic mechanisms involved [135].

## 6. A New Way Forward: Modulating the GM with Precision Probiotics

As previously mentioned, the Notch pathway is deeply involved in immune and inflammatory processes, modulating the secretion of multiple inflammatory cytokines, such as IL-1β, IL-6, TNF-α, IL-18, and IFN-γ [82]. It plays a complex role in inflammation and can induce the activation of immune cells, such as T lymphocytes and macrophages, leading these cells to secrete these factors. The ratio of these molecules can dictate the intensity and duration of the inflammatory response.

One promising strategy is GM modulation with specific probiotic strains. For example, *Bifidobacterium longum* ES1 has demonstrated anti-inflammatory effects on intestinal diseases in various models, e.g., in vitro and animal models, attenuating both spontaneous and chemically induced colitis via the regulation of cytokines or a targeted induction of immune regulation mechanisms [171].

Sichetti et al. simulated the intestinal epithelial barrier function in vitro with *B. longum* and macrophages, proving that probiotics significantly influenced the production of IL-10. At the same time, the secretory IL-1β and IL-6 levels were inhibited by 70% and 80%, respectively [172].

Similarly, Singh et al. reported that *B. longum* can modulate cellular signaling pathways, leading to decreased levels of proinflammatory cytokines, such as IL-1β, IL-6, and IL-8. Furthermore, it alleviates the DSS-induced alteration of the in vitro epithelial barrier and regulates the inflammatory response [173].

Another emerging player is *Clostridium butyricum*. Its anti-inflammatory effects and ability to stimulate Treg responses are closely linked to its capacity to increase IL-10 levels [174]. In a murine colitis model, Hayashi et al. documented the essential role of macrophage-derived IL-10 in mediating the protective effect of CBM 588, as its depletion abrogated the protective effect of the strain. *C. butyricum* induces colonic macrophage IL-10 production through TLR2 activation [175]. CBM 588 is known to change TLR2 signaling and inhibit pathogen-induced inflammation and apoptosis in vitro [176]. *C. butyricum* induces an anti-inflammatory Treg response, mediating IL-10 and TGF-β, possibly via the activation of a TLR2-dependent pathway [177].

*Streptococcus salivarius* is particularly noteworthy in the context of recurrent pharyngitis or GM oralization. This bacterium colonizes the human oral cavity within days of birth and remains one of the most abundant oral commensals. It has also been found in the stomach and jejunum, which suggests that it may play an important role in the ecology of both the oral cavity and gastrointestinal tracts [178]. In different studies focusing on gingival, bronchial, pharyngeal, and intestinal epithelial mucosa, suppression of NF-κB activation and IL-8 production was observed following stimulation with flagellin from other microorganisms [179]. Macdonald et al. demonstrated that *Streptococcus salivarius* can prevent the immune activation by periodontal disease pathogens [180]. In gingival fibroblasts stimulated with the pathogens *Porphyromonas gingivalis* and *Fusobacterium nucleatum,* administration of *S. salivarius* K12 and M18 has an effect on the production of IL-6: when both probiotics were administered either with pathogens or after pre-treatment of fibroblasts, a decrease in cytokine production was recorded.

Finally, research by Kim et al. showed that threonyl-tRNA synthetase (AmTARS) secreted by purified *Akkermansia muciniphila* ARSs stimulates M2 macrophage polarization and regulates IL-10 [181]. The relevant pathways include MAPK and PI3K/AKT, and these pathways ultimately converge upon the cAMP response element-binding protein (CREB). This pathway enhances IL-10 synthesis and suppresses the key inflammatory mediator, NF-kB. Furthermore, a pasteurized strain of *A. muciniphila Muc^T^* and its outer membrane protein Amuc-1100 have been shown to downregulate collagen and proinflammatory cytokines in murine CRC, and decrease mRNA expression of proinflammatory cytokines in DSS-induced colitis [182]. Live *A. muciniphila* had no effects on TNF or IL-1b expression in inflamed colon tissue but enhanced IL-10 release, in line with an anti-inflammatory profile. Proinflammatory markers were elevated by heat-killed *A. muciniphila* and *E. coli*. *Akkermansia* showed a positive correlation with IL-10 and a negative correlation with IL-6 and TNF [183]. Higher Tlr2 and Smad3 expression induced particularly with live *A. muciniphila* points towards an immune modulatory function, potentially through the TGF-β pathway [184].

Moreover, Molaaghaee-Rouzbahani et al. explored the anti-inflammatory properties of *Akkermansia muciniphila* in gliadin-stimulated macrophages (MQs) [185]. Gliadin triggered a proinflammatory M1 phenotype with TNF-α and IL-6. *A. muciniphila* pre- and post-treatment induced the shift of MQs to an anti-inflammatory M2 phenotype and reduced the proinflammatory markers (IL-6, TNF-α) while boosting the level of anti-inflammatory markers (IL-10, TGF-β). However, *Akkermansia muciniphila* is currently only available as a postbiotic [186].

Other well-studied probiotics include *Lactobacillus rhamnosus GG*, which acts through the upregulation of anti-inflammatory cytokines, including IL-4 and IL-10, along with the downregulation of inflammatory cytokines, including IL-1, IL-6, and TNF-α [187]. Multiple members of the Lactobacillus and Bifidobacterium families have shown potential organ-targeting ability in terms of therapy, supporting the need for further studies. Moreover, if the right combination of Lactobacillus and Bifidobacterium species is exhibited to reduce symptoms in multiple areas of the body, it could reduce the need for multiple doses and/or medications [187].

A meta-analysis conducted by McLoughlin et al. evaluated 29 prebiotic studies [188]. Fourteen of these studies, accounting for 48%, reported a decline in inflammatory markers. More specifically, among the 13 studies that focused on oligosaccharide prebiotics, nine (approximately 69%) observed a substantial reduction in at least one inflammatory biomarker (namely TNF-α, IL-6, CRP, or interferon-γ) when compared to control groups. Conversely, two crossover studies involving healthy participants indicated an increase in inflammatory markers (CRP, TNF-α, and IL-6), while two additional studies targeting populations with gastrointestinal disorders, such as Crohn’s disease and acute diarrhea, did not find significant effects on inflammation.

In the context of polysaccharide prebiotics, only two of the ten studies (20%) revealed a noteworthy reduction in inflammation when compared to their control or baseline measurements, with the remaining eight studies indicating no evidence of an anti-inflammatory impact [188]. Of the two studies investigating resistant starch supplementation, the one of Aliasgharzadeh et al. reported a meaningful decrease in TNF-α and IL-6 levels [189]. Furthermore, a high soluble fiber diet (10.7 g daily) was associated with a significant reduction in CRP compared to a low soluble fiber diet (2.5 g daily). A cross-sectional study conducted by Ma et al. established an inverse relationship between soluble fiber consumption and systemic inflammation (IL-6 and TNF-α-R2) [190]. However, numerous papers are contradictory. The existing data on efficacy are not yet robust enough to satisfy the demanding criteria the United States Preventive Services Task Force uses for a preventive recommendation [191]. But this does not mean probiotics are not effective in healthy people—it means there are not enough data to support a population-wide recommendation by the United States Preventive Services Task Force. The research around probiotics is still relatively young, and a lack of definitive evidence of effectiveness should not be interpreted as evidence that they do not work. Indeed, probiotics are the only reasonable option in certain circumstances based on available efficacy and safety data [192]. However, the evidence is not strong enough to justify broad preventive recommendations for the general population.

Yet, probiotics can also have side effects. Recent evidence has associated specific probiotic strains with serious infections, potentially resulting in sepsis, bacteremia, fungemia, endocarditis, and other opportunistic infections [193]. Besides that, the production of cytokines, such as IL-1β, IL-6, IFN, and TNF-α, associated with probiotics may elicit an excessive immune response, resulting in an inflammatory response or autoimmune diseases [194]. These risks warrant a reconsideration of the use of probiotics in patients with autoimmune diseases, particularly in people who are at higher risk.

In addition, probiotic interventions could lead to sensitization in susceptible children and adults. Reduced microbial diversity-associated allergic disorders have been documented in early childhood, corresponding with lower levels of lactobacilli and bifidobacteria [195].

The diversity in strain types, dosages, and study populations needs further evaluation for effective and consistent probiotic therapy. Various uncertainties and challenges still exist; however, probiotics have proven to be a potentially safe and effective adjunct treatment option for a wide range of disorders, improving the quality of life of patients and helping them regulate their gut health [196], Further research is needed to better elucidate mechanisms of action, formulations, and strain responses before utilizing probiotics for therapeutic gains in gastrointestinal health.

Table 2 summarizes the immunomodulatory effects of the mentioned probiotics.

Modulating the GM may have beneficial effects in autoinflammatory diseases through their influence on the immune system and the Notch signaling pathway, but the risks, uncertainties, and paradoxical data must also be considered, too.

Findings from animal models do not always translate directly to humans. Furthermore, human responses to probiotics are extremely heterogeneous based on age, gender, genetics, diet, gut microbiota composition, and health status [197]. Additionally, probiotic interventions should not be viewed in isolation, but as part of a comprehensive system that includes general dietary and lifestyle factors. Diet quality, exercise, stress, and sleep patterns are among the factors that play a very important role in maintaining gut health and microbial balance, thereby influencing the efficacy of probiotics [198].

## 7. Conclusions

Notch signaling is a key regulator of inflammation in several autoinflammatory diseases. Its wide-ranging implications suggest that mutations affecting Notch regulatory mechanisms are highly context-dependent across different diseases, preventing a one-size-fits-all therapeutic approach. Insights into the pathways involved in Notch activation and their interplay with other signaling networks will provide important information to delineate the unique clinical findings of each disease.

Such effects indicate that Notch signaling inhibition could provide an effective therapeutic target. However, the variability in therapeutic responses highlight the need to study differences in Notch regulation at the tissue- or cell-type level to guide the development of improved targeted inhibitors.

A key component is the interaction between Notch signaling and the GM: evidence supports the idea that GM could regulate Notch activity, and that Notch could in turn influence GM composition and contribute to gut health and inflammation. Strain-specific probiotics may not only restore the GM balance, but also enhance Notch signaling regulation, resulting in a synergistic therapeutic strategy.

Nevertheless, large-scale studies are essential to establish standardized microbiota assessment protocols and determine the effective probiotics for each patient. GM composition and immune response vary considerably from person to person, arguing for a personalized approach in therapy.

Notably, the knowledge gained about Notch signaling and the GM may offer bases for new therapeutic strategies against autoinflammatory diseases. The additive interplay of Notch inhibitors and probiotics can propose a transient semi-meditated therapeutic strategy intending to ameliorate clinical outcomes, as well as enhance the quality of life in patients. Nevertheless, discrepancies remain in the literature regarding the link between the Notch signaling pathway and the GM in autoinflammatory diseases. However, the overall effects of all GM variations on Notch signaling in different disease contexts, as well as the specific underlying mechanisms responsible for this interaction, are not entirely known. Moreover, although some studies imply that targeted probiotics might modulate the inflammatory response via Notch, clinical evidence is scarce and fragmented. Subsequent studies must concentrate on pinpointing exact molecular targets and clinical trials of therapeutic approaches based on the modulation of Notch and the GM to pave the way for increasingly personalized and effective strategies in the handling of autoinflammatory diseases.

## Figures and Tables

**Figure 1 biomedicines-13-00768-f001:**
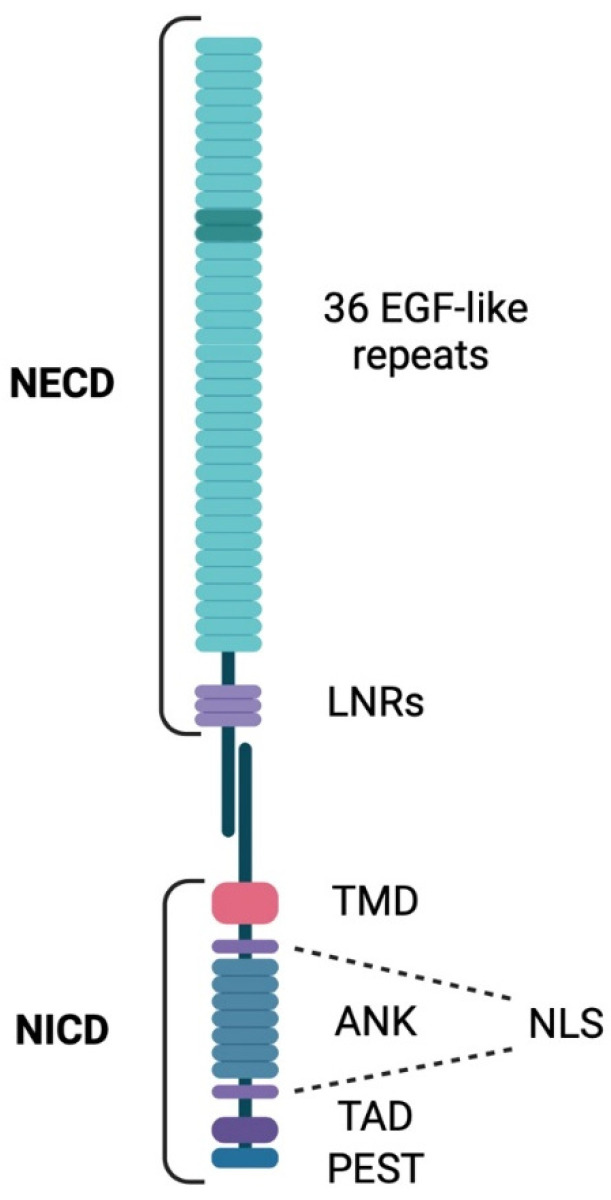
Notch1 receptor includes the extracellular (NECD) and intracellular (NICD) domains. The NECD consists of 36 epidermal growth factor (EGF)-like repeat and LIN-12/Notch repeat (LNRS) regions. NECD and NICD are linked by the transmembrane domain (TMD). NICD contains ankyrin repeats (ANKs), a nuclear localization domain (NLS), transcription activation domain (TAD), and proline-glutamic acid-serine-threonine (PEST) domain (created with BioRender.com, accessed on 11 October 2024).

**Figure 2 biomedicines-13-00768-f002:**
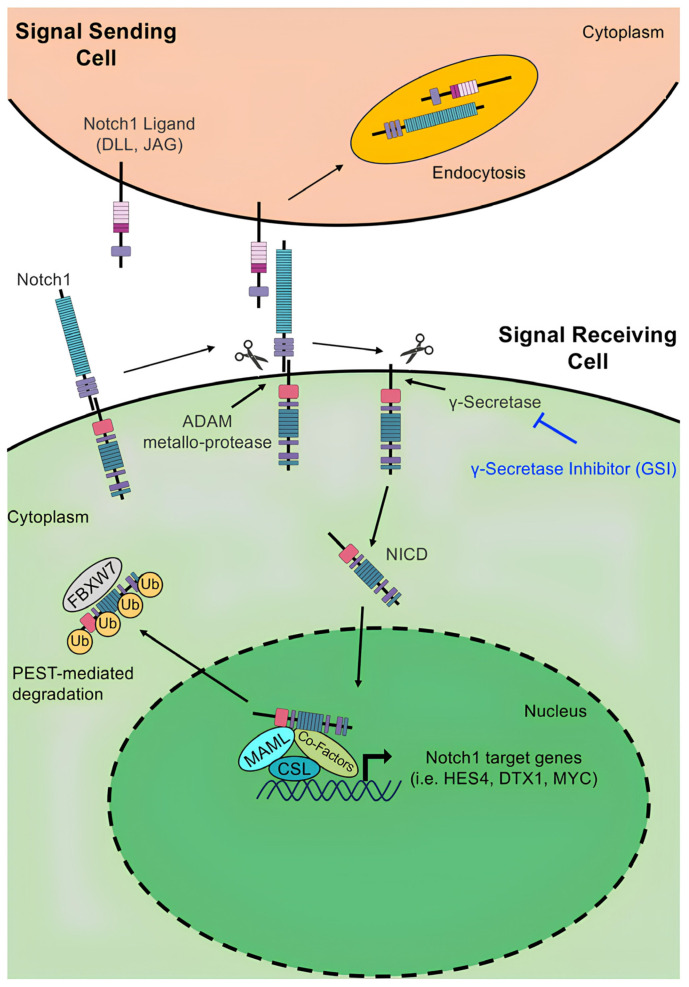
Notch1 pathway. Notch1 ligand (DLL, JAG) on the sending cell (Signal Sending Cell). The Notch1 receptor is activated on the receiving cell (Signal Receiving Cell), undergoes endocytosis, and is processed by ADAM metalloprotease and γ-Secretase. The release of NICD into the cytoplasm allows it to translocate into the nucleus where it binds to CSL, MAML, and co-factors to modulate the expression of Notch1 target genes (i.e., HES4, DTX1, MYC) in cooperation with other transcriptional regulators. FBXW7 and PEST mediate NICD degradation.

**Figure 3 biomedicines-13-00768-f003:**
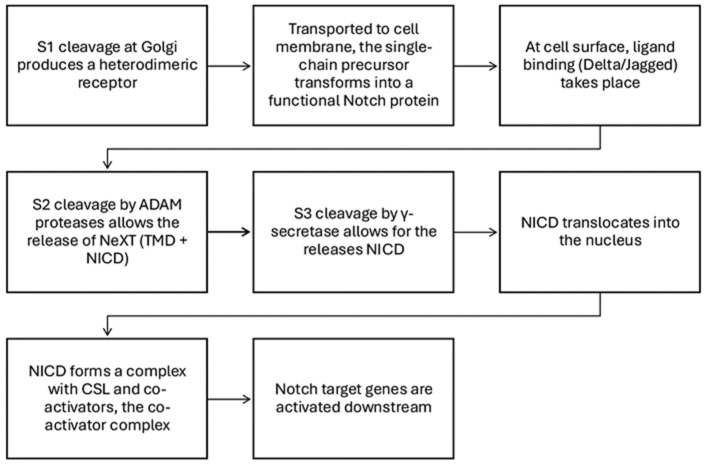
In the flowchart, the various steps of the Notch activation process are reported.

**Figure 4 biomedicines-13-00768-f004:**
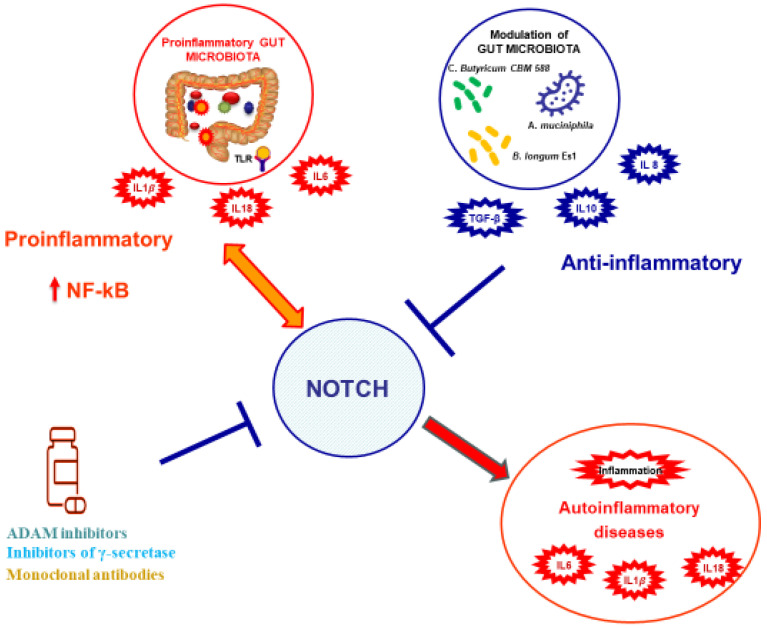
GM is a key player in inflammation and can contribute to the onset of autoinflammatory diseases. Proinflammatory GM release substances that are able to activate several mediators of inflammation, such as interleukins. On the other side, some beneficial bacteria can secrete compounds that help to create an anti-inflammatory environment. Notch is a main player in this game. When Notch is activated, inflammation increases, and autoinflammatory diseases may develop. Some drugs can downregulate the Notch activity, reducing inflammation. In this way, Notch represents a potential target to treat autoinflammatory diseases.

**Table 2 biomedicines-13-00768-t002:** Probiotics and their respective strain-specific immunomodulatory properties.

Probiotic	Immunomodulatory Effect	References
Bifidobacterium Longum ES1	- Inducing IL-10 production - Reduction in IL-1β and IL-6 IL-8 levels	[171,172,173]
Clostridium butyricum CBM 588	- Stimulating Treg response - Modifying TLR2 signaling to inhibit pathogen-induced inflammation - Inducing IL-10 and TGF-β	[174,176,177]
Streptococcus salivarius	- Inhibition of NF-κB activation and IL-8 production	[179]
Akkermansia muciniphila (currently available on the market only as a postbiotic)	- Inhibition of NF-κB activation - Enhanced IL-10 - Modulation of TGF-β - Reduction IL-6 and TNF-α levels	[181,183,184,185]
Lactobacillus rhamnosus GG	- Upregulation of anti-inflammatory cytokines IL-4 and IL-10 - Downregulation of inflammatory cytokines IL-1, IL-6, and TNF-α	[187]

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
