# Peer review of "The Role of Notch Signaling and Gut Microbiota in Autoinflammatory Diseases: Mechanisms and Future Views"

_biomedicines, 2025, doi:10.3390/biomedicines13040768_

Round 1

Reviewer 1 Report

Comments and Suggestions for Authors

The manuscript aims to narrate notch signaling, anti-inflammatory disease, and how gut microbiota might be related to them. Starting with an introduction, followed by notch signaling pathways and gut microbiota-related inflammatory diseases. Finally, suggesting the possibility of using probiotics (not gut microbiota) to modulate Notch signaling, resulting in the treatment of related diseases. 

Major comments:

Introduction: An introduction to GM is missing 

An introduction to "Autoinflammatory disease" is missing (See part 2)

There is a gap between line 73 and 74 to explain what a Gut microbiota is (See part 3) 

Reference 12 does not mention helicobacter pylori and does not match this: "Similarly, Helicobacter pylori activation of Nucleotide- 82 binding Leucine-rich Repeat-containing 12 (NLRP12) suppresses Notch signaling and 83 leads to decreased intestinal epithelial cell production of inflammatory chemokines, such 84 as Monocyte chemoattractant protein-1 (MCP-1) and Macrophage Inflammatory Pro- 85 teins-1α (MIP-1α)".

Part 2. Line 204 : A definition or explanation of "autoinflammatory disease" should be placed in the introduction.

Part 3. The first paragraph ( is an introduction to the Gut microbiota and should go to the introduction. 

**There are disconnections between some parts of the manuscript. For example, there is a gap of explanation from lines 343 to 352.  

**Parts 3 and 4 titles are the same. 

Line 237: "there are six main phyla (firmicutes, bacteroidetes, actinobacteria, proteobacteria, fusobacteria, and verrucomicrobia) and their equilibrium has serious consequences on human healthcare". This statement should be clear. Do these phyla have serious consequences? This statement misleads the concept of Microbiome. 

**Figure 1: This figure might be helpful, but it is not essential and will not add significant value to the manuscript. Please provide a figure explaining part 2. Notch signaling pathway

**Figure 2: More comprehensive details are needed to explain the content of the manuscript. The authors should improve this figure. For example, does gut microbiota only inhibits Notch signaling or it can also increase it? The confusion between "Gut Microbiota" and "Probiotics" in the whole manuscript should be fixed. As another example: Akkermancia muciniphila is a member of gut microbiota, but it is not considered as a probiotic yet (Table 2). 

*Table 1. Consider rewriting the legend "The primary target of molecules active in the Notch pathway". 

**Table 2. Akkermancia muciniphila is not a probiotic yet. It is recommended to review other mentioned microorganisms in this table and consider if they are Probiotics or just a member of gut microbiota. There should be many more published papers showing other members of the gut microbiota with anti-inflammatory potential and some of them are used as probiotics in the market. 

Part 7. How does the title "Patent" relate to the content of this part?

It is essential to reconsider an extensive review of all parts of this review manuscript.

Notch signaling pathway.

Comments on the Quality of English Language

It is highly recommended to carefully review the language of the manuscript. 

Author Response

Reviewer 1
The manuscript aims to narrate notch signaling, anti-inflammatory disease, and how gut microbiota
might be related to them. Starting with an introduction, followed by notch signaling pathways and gut
microbiota-related inflammatory diseases. Finally, suggesting the possibility of using probiotics (not
gut microbiota) to modulate Notch signaling, resulting in the treatment of related diseases. 
Major comments:

ï‚· Introduction: An introduction to GM is missing 
We have included an introduction to GM, as suggested.
ï‚· An introduction to "Autoinflammatory disease" is missing (See part 2)
Thanks for the suggestion. We have included an introduction to autoinflammatory diseases.
ï‚· There is a gap between line 73 and 74 to explain what a Gut microbiota is (See part 3) 
We rewrote this sentence to solve the gap
ï‚· Reference 12 does not mention helicobacter pylori and does not match this: "Similarly,
Helicobacter pylori activation of Nucleotide- 82 binding Leucine-rich Repeat-containing 12
(NLRP12) suppresses Notch signaling and 83 leads to decreased intestinal epithelial cell
production of inflammatory chemokines, such 84 as Monocyte chemoattractant protein-1
(MCP-1) and Macrophage Inflammatory Pro- 85 teins-1α (MIP-1α)".
We have corrected the reference.
ï‚· Part 2. Line 204: A definition or explanation of "autoinflammatory disease" should be placed
in the introduction.
The definition of autoinflammatory diseases was included in the introduction, as suggested.
ï‚· Part 3. The first paragraph (is an introduction to the Gut microbiota and should go to the
introduction. 
We have modified the text, as suggested.

ï‚· **There are disconnections between some parts of the manuscript. For example, there is a gap
of explanation from lines 343 to 352.  
We thank the reviewer and we have modified the manuscript, as suggested
ï‚· **Parts 3 and 4 titles are the same. 
We corrected the mistake
ï‚· Line 237: "there are six main phyla (firmicutes, bacteroidetes, actinobacteria, proteobacteria,
fusobacteria, and verrucomicrobia) and their equilibrium has serious consequences on human
healthcare". This statement should be clear. Do these phyla have serious consequences? This
statement misleads the concept of Microbiome. 
We have modified the sentence so that the meaning is clearer.
ï‚· **Figure 1: This figure might be helpful, but it is not essential and will not add significant
value to the manuscript. Please provide a figure explaining part 2. Notch signaling pathway
We have modified the figure 1 to explain the Notch signaling pathway, according to the suggestion.
ï‚· **Figure 2: More comprehensive details are needed to explain the content of the manuscript.
The authors should improve this figure. For example, does gut microbiota only inhibits Notch
signaling or it can also increase it? The confusion between "Gut Microbiota" and "Probiotics"
in the whole manuscript should be fixed. As another example: Akkermancia muciniphila is a
member of gut microbiota, but it is not considered as a probiotic yet (Table 2). 
We have modified the figure 2 and rewritten the caption, according to the suggestion.
ï‚· *Table 1. Consider rewriting the legend "The primary target of molecules active in the Notch
pathway". 
We have rewritten the legend, as requested.
ï‚· **Table 2. Akkermancia muciniphila is not a probiotic yet. It is recommended to review other
mentioned microorganisms in this table and consider if they are Probiotics or just a member of
gut microbiota. There should be many more published papers showing other members of the
gut microbiota with anti-inflammatory potential and some of them are used as probiotics in the
market. 
We made it clear that Akkermansia muciniphila is only marketed as a postbiotic. We have also
checked that the other microorganisms mentioned are probiotics.
ï‚· Part 7. How does the title "Patent" relate to the content of this part?
It was a mistake, we corrected it.

Reviewer 2 Report

Comments and Suggestions for Authors

This great review examines the effect of the notch pathway on the microbiome in inflammation. The article is structured and comprehensive, along with an informative figure and tables. I have some comments:

- A conclusion should be added, identifying a gap in the current literature.

- Section 4: a part concerning Irritable Bowel Syndrome (IBS) should be added.

- Section 6: some additions concerning prebiotics should be made.

Author Response

This great review examines the effect of the notch pathway on the microbiome in inflammation. The
article is structured and comprehensive, along with an informative figure and tables. I have some
comments:

ï‚· A conclusion should be added, identifying a gap in the current literature.
We have rewritten the conclusion, discussing the current knowledge of the literature

ï‚· Section 4: a part concerning Irritable Bowel Syndrome (IBS) should be added.
We have added sentences on IBS, as requested, even though it is not considered an autoinflammatory
disease
ï‚· Section 6: some additions concerning prebiotics should be made.
We have modified the text, as suggested.

Reviewer 3 Report

Comments and Suggestions for Authors

The review of Giambra et al. aimed to summarize current research on the interplay between Notch signaling, gut microbiota, and autoinflammatory diseases to provide insights into potential new therapeutic approaches. Overall, the manuscript provides an insightful review of the current state of research in the role of Notch signaling and gut microbiota in autoinflammatory diseases. However, several areas require attention to enhance the flow, coherence, logical progression, structure and organization, and clarity of the information presented. The inclusion of more figures or schematic diagrams summarizing the complex interactions discussed would significantly enhance the manuscript's accessibility and comprehension. Below are some suggestions for improvements in the text:

Abstract: The abstract is fragmented into several paragraphs (lines 14-36), each discussing different aspects of the research on Notch signaling, Gut Microbiota (GM), and their roles in autoinflammatory diseases. This fragmented approach can disrupt the flow and coherence, making it challenging for readers to grasp the study's essence quickly. For improvement, the abstract's content should be condensed and organized into a single paragraph.

Introduction: The introduction effectively sets the stage for the discussion by providing background information on Notch signaling and its importance in cell fate determination and immune regulation. The linkage to autoinflammatory diseases is clearly stated, providing a rationale for the study. I suggest to provide some background on autoinflammatory diseases themselves, including prevalence, impact, and current gaps in treatment or understanding. This would help contextualize the importance of exploring Notch signaling and gut microbiota in these diseases.

Line 70: The transition between the introduction of Notch signaling and its interaction with GM in the context of autoinflammatory diseases is abrupt. The text jumps from explaining the Notch signaling pathway directly to its modulation by GM without a smooth transition. Introduce a bridging paragraph that outlines the significance of studying the interaction between Notch signaling and GM in the context of autoinflammatory diseases. This could provide a smoother transition and prepare the reader for the detailed discussions that follow.

Notch Signaling Pathway: If possible, add figures or models to summarize these complex interactions to aid reader comprehension. The text abruptly shifts from detailing Notch signaling's impact on immune cells to describing the GM's composition (Section 3). Insert a paragraph that links the discussion on Notch signaling in immune responses to the introduction of GM, emphasizing the importance of GM in modulating these responses and setting the stage for a discussion on their interplay. In addition, the description of Notch signaling pathway components and their functions is dense and could be overwhelming. Simplify explanations and introduce concepts gradually. I suggest the authors to use diagrams to visually represent complex pathways and interactions, and include definitions or brief explanations of technical terms when they first appear.

Section 3 and 4: They have the same title - Notch pathway, gut microbiota and autoinflammation. However, the content of these sections differs. Section 3 (the first occurrence) provides a general overview of the relationship between the Notch pathway, gut microbiota, and autoinflammation. It discusses how the gut microbiota can modulate Notch signaling and vice versa, and how this interaction affects inflammatory processes. Section 4 (the second occurrence) goes into more specific details about the role of Notch signaling and gut microbiota in particular autoinflammatory diseases. It covers conditions such as Behçet's disease, inflammatory bowel diseases (IBD), and cryopyrin-associated periodic syndromes (CAPS). The repetition of the section title appears to be an error, as the content is distinct in each section

Section 4 provides a comprehensive review of the relationship between the Notch pathway, gut microbiota (GM), and autoinflammation. While the detailed exploration of diseases like Pyoderma gangrenosum, Behçet’s disease, and IBD provides valuable insights, the dense medical details may detract from the broader narrative on the Notch pathway and GM's roles. Balance the detailed disease-specific information with insights that tie back to the Notch pathway and GM more explicitly. For example, after discussing a disease, summarize the implications for understanding the Notch-GM-autoinflammation nexus. Also, there is repetitive information across the diseases discussed, particularly regarding the mechanisms of Notch signaling and the role of GM. I suggest to consolidate repetitive descriptions of Notch signaling and GM's roles into a single, concise section that precedes the disease-specific discussions. This approach would avoid redundancy and allow for a smoother narrative flow.

Notch-targeted drugs and future perspectives: The therapeutic section is optimistic but lacks a critical analysis of the challenges and limitations faced by current therapeutic approaches. Please, try to provide a more balanced view, including the potential side effects of manipulating the Notch pathway and the gut microbiota, would provide a more comprehensive overview.

A new way forward: modulating GM with precision probiotics: The section predominantly focuses on the benefits of modulating the gut microbiota with probiotics, without adequately addressing potential risks, limitations, or conflicting evidence in the field. A more balanced view, including the complexity of host-microbe interactions, potential for adverse effects, and instances where probiotic interventions have failed to show benefits, would provide a more nuanced perspective. Please, specify the strains of Akkermansia muciniphila described in the text.

Section 7: The designation of this section as "Patents" is not consistent with its content. This section actually contains a conclusion or summary of the review rather than information about patents. 

Author Response

The review of Giambra et al. aimed to summarize current research on the interplay between Notch
signaling, gut microbiota, and autoinflammatory diseases to provide insights into potential new
therapeutic approaches. Overall, the manuscript provides an insightful review of the current state of
research in the role of Notch signaling and gut microbiota in autoinflammatory diseases. However,
several areas require attention to enhance the flow, coherence, logical progression, structure and
organization, and clarity of the information presented. The inclusion of more figures or schematic
diagrams summarizing the complex interactions discussed would significantly enhance the
manuscript's accessibility and comprehension. Below are some suggestions for improvements in the
text:

ï‚· Abstract: The abstract is fragmented into several paragraphs (lines 14-36), each discussing
different aspects of the research on Notch signaling, Gut Microbiota (GM), and their roles in
autoinflammatory diseases. This fragmented approach can disrupt the flow and coherence,
making it challenging for readers to grasp the study's essence quickly. For improvement, the
abstract's content should be condensed and organized into a single paragraph.
Thank you for the comment, we have modified the abstract following your suggestions.

ï‚· Introduction: The introduction effectively sets the stage for the discussion by providing
background information on Notch signaling and its importance in cell fate determination and
immune regulation. The linkage to autoinflammatory diseases is clearly stated, providing a
rationale for the study. I suggest to provide some background on autoinflammatory diseases
themselves, including prevalence, impact, and current gaps in treatment or understanding.

This would help contextualize the importance of exploring Notch signaling and gut microbiota
in these diseases.
We have included a section on autoinflammatory diseases in the introduction according to the
suggestions.

ï‚· Line 70: The transition between the introduction of Notch signaling and its interaction with
GM in the context of autoinflammatory diseases is abrupt. The text jumps from explaining the
Notch signaling pathway directly to its modulation by GM without a smooth transition.
Introduce a bridging paragraph that outlines the significance of studying the interaction
between Notch signaling and GM in the context of autoinflammatory diseases. This could
provide a smoother transition and prepare the reader for the detailed discussions that follow.
We have rewritten the text, as requested

ï‚· Notch Signaling Pathway: If possible, add figures or models to summarize these complex
interactions to aid reader comprehension. The text abruptly shifts from detailing Notch
signaling's impact on immune cells to describing the GM's composition (Section 3). Insert a
paragraph that links the discussion on Notch signaling in immune responses to the
introduction of GM, emphasizing the importance of GM in modulating these responses and
setting the stage for a discussion on their interplay. In addition, the description of Notch
signaling pathway components and their functions is dense and could be overwhelming.
Simplify explanations and introduce concepts gradually. I suggest the authors to use diagrams
to visually represent complex pathways and interactions, and include definitions or brief
explanations of technical terms when they first appear.
Thank you for your comments. We have added a paragraph introducing the GM and we have
simplified the description of the Notch signalling pathway. We have added the figure 2 and a
flowchart, which further exemplifies the pathway.

ï‚· Section 3 and 4: They have the same title - Notch pathway, gut microbiota and
autoinflammation. However, the content of these sections differs. Section 3 (the first
occurrence) provides a general overview of the relationship between the Notch pathway, gut
microbiota, and autoinflammation. It discusses how the gut microbiota can modulate Notch
signaling and vice versa, and how this interaction affects inflammatory processes. Section 4
(the second occurrence) goes into more specific details about the role of Notch signaling and
gut microbiota in particular autoinflammatory diseases. It covers conditions such as Behçet's
disease, inflammatory bowel diseases (IBD), and cryopyrin-associated periodic syndromes
(CAPS). The repetition of the section title appears to be an error, as the content is distinct in
each section

Section 4 provides a comprehensive review of the relationship between the Notch pathway, gut
microbiota (GM), and autoinflammation. While the detailed exploration of diseases like
Pyoderma gangrenosum, Behçet’s disease, and IBD provides valuable insights, the dense
medical details may detract from the broader narrative on the Notch pathway and GM's roles.
Balance the detailed disease-specific information with insights that tie back to the Notch
pathway and GM more explicitly. For example, after discussing a disease, summarize the
implications for understanding the Notch-GM-autoinflammation nexus. Also, there is
repetitive information across the diseases discussed, particularly regarding the mechanisms of
Notch signaling and the role of GM. I suggest to consolidate repetitive descriptions of Notch
signaling and GM's roles into a single, concise section that precedes the disease-specific
discussions. This approach would avoid redundancy and allow for a smoother narrative flow.
Thank you for your comment, we have modified the title of section three. As for section four, we
have followed your suggestion and added a more general section describing the most common
mechanisms of Notch signalling and modified the single sections of the diseases.

ï‚· Notch-targeted drugs and future perspectives: The therapeutic section is optimistic but lacks a
critical analysis of the challenges and limitations faced by current therapeutic approaches.
Please, try to provide a more balanced view, including the potential side effects of manipulating
the Notch pathway and the gut microbiota, would provide a more comprehensive overview.
We added the adverse effects of various therapeutic approaches to the Notch pathway and
modulation of the gut microbiota

ï‚· A new way forward: modulating GM with precision probiotics: The section predominantly
focuses on the benefits of modulating the gut microbiota with probiotics, without adequately
addressing potential risks, limitations, or conflicting evidence in the field. A more balanced
view, including the complexity of host-microbe interactions, potential for adverse effects, and
instances where probiotic interventions have failed to show benefits, would provide a more
nuanced perspective. Please, specify the strains of Akkermansia muciniphila described in the
text.
We added the possible adverse effects of microbiota modulation and specified the strain of
Akkermansia muciniphila as required.

ï‚· Section 7: The designation of this section as "Patents" is not consistent with its content. This
section actually contains a conclusion or summary of the review rather than information about
patents. 
Thank you, we have switched the title to “Conclusions”

Round 2

Reviewer 1 Report

Comments and Suggestions for Authors

With a random check, some references are not appropriately cited—for example, ref 44, page 10, line 31.  Or reference 193 is not an appropriate reference to state that the A. muciniphila is a postbiotic.  It is essential to check all the references if they are cited appropriately. 

Figure 1 and Figure 2 legends are the same.

Figure 2 shows the maturation and activation of Notch proteins, but it does not show the signaling pathway. 

Figure 3. There is no major revision. Only a double arrow was added to this figure. Authors rewrote the caption with the details that are not present in the figure. These details should go to the body of the manuscript. 

Author Response

With a random check, some references are not appropriately cited—for example, ref 44, page 10, line 31.  Or reference 193 is not an appropriate reference to state that the A. muciniphila is a postbiotic.  It is essential to check all the references if they are cited appropriately. 

  • We thank the reviewer. We have checked all the references and modified them accordingly.

Figure 1 and Figure 2 legends are the same.

  • We have corrected the mistake.

Figure 2 shows the maturation and activation of Notch proteins, but it does not show the signaling pathway. 

  • We thank the reviewer. We have modified the figure 2.

Figure 3. There is no major revision. Only a double arrow was added to this figure. Authors rewrote the caption with the details that are not present in the figure. These details should go to the body of the manuscript. 

  • We have modified the figure 3 and rewritten the caption. Thank you for the suggestion.

Reviewer 3 Report

Comments and Suggestions for Authors

The authors have adequately addressed all the comments. The manuscript may be accepted for publication.

Author Response

The authors have adequately addressed all the comments. The manuscript may be accepted for publication.

  • We thank the reviewer.